# Cardiac Protective Effect of Kirenol against Doxorubicin-Induced Cardiac Hypertrophy in H9c2 Cells through Nrf2 Signaling via PI3K/AKT Pathways

**DOI:** 10.3390/ijms22063269

**Published:** 2021-03-23

**Authors:** Abdullah M. Alzahrani, Peramaiyan Rajendran, Vishnu Priya Veeraraghavan, Hamza Hanieh

**Affiliations:** 1Department of Biological Sciences, College of Science, King Faisal University, Al Ahsa 31982, Saudi Arabia; aalzahra@kfu.edu.sa; 2Department of Biochemistry, Saveetha Dental College, Saveetha Institute of Medical and Technical Sciences, Saveetha University, Chennai 600 077, India; vishnupriya@saveetha.com; 3Department of Medical Analysis, Al-Hussein Bin Talal University, Ma’an 71111, Jordan; hhanieh@ahu.edu.jo

**Keywords:** doxorubicin, kirenol, Nrf2, H9c2, hypertrophy, cardiotoxicity

## Abstract

Kirenol (KRL) is a biologically active substance extracted from Herba Siegesbeckiae. This natural type of diterpenoid has been widely adopted for its important anti-inflammatory and anti-rheumatic properties. Despite several studies claiming the benefits of KRL, its cardiac effects have not yet been clarified. Cardiotoxicity remains a key concern associated with the long-term administration of doxorubicin (DOX). The generation of reactive oxygen species (ROS) causes oxidative stress, significantly contributing to DOX-induced cardiac damage. The purpose of the current study is to investigate the cardio-protective effects of KRL against apoptosis in H9c2 cells induced by DOX. The analysis of cellular apoptosis was performed using the terminal deoxynucleotidyl transferase dUTP nick-end labeling (TUNEL) staining assay and measuring the modulation in the expression levels of proteins involved in apoptosis and Nrf2 signaling, the oxidative stress markers. Furthermore, Western blotting was used to determine cell survival. KRL treatment, with Nrf2 upregulation and activation, accompanied by activation of PI3K/AKT, could prevent the administration of DOX to induce cardiac oxidative stress, remodeling, and other effects. Additionally, the diterpenoid enhanced the activation of Bcl2 and Bcl-xL, while suppressing apoptosis marker proteins. As a result, KRL is considered a potential agent against hypertrophy resulting from cardiac deterioration. The study results show that KRL not only activates the IGF-IR-dependent p-PI3K/p-AKT and Nrf2 signaling pathway, but also suppresses caspase-dependent apoptosis.

## 1. Introduction

With significant advancements in cancer screening and the rapid development of anticancer strategies, cancer treatment has markedly improved in recent times. Subsequently, the number of cancer survivors continues to be on the rise with each passing year. However, some of these treatments and drugs may cause short- or long-term side effects. Chemotherapeutic agents, in particular, can induce complications in patients with cancer that delay or interrupt the treatment. These complications may affect the quality of life and increase morbidity and mortality associated with cancer treatment [1]. Heart failure is one of the most common chemotherapy-induced complications. The American Society of Clinical Oncology (ASCO) guidelines reported that patients aged over 65 years with underlying medical conditions such as arterial hypertension, coronary artery disease, and dyslipidemia are at a high risk of suffering from chemotherapy-induced cardiotoxicity [2,3,4]. One of the most commonly used chemotherapeutic drugs is an anthracycline antibiotic—doxorubicin (DOX)—derived from *Streptomyces*. It is considered to be effective for various types of cancers, including leukemia, solid tumors, soft-tissue sarcomas, and breast cancer [5]. However, DOX is dose-dependent and has toxic effects on cardiomyocytes, which remains the main concern among patients and healthcare professionals. Additionally, when DOX exceeds the cumulative dosage of 400–700 mg/m^2^ and 300 mg/m^2^ in adults and children respectively, it can cause congestive heart failure (CHF) [5,6]. A hypothesis based on the formation of free radicals, inducing oxidative stress, has been widely investigated for DOX-induced cardiotoxicity [7,8,9].

Oxidative stress is one of the leading causes of DOX-induced cardiotoxicity. It is involved in a wide range of processes such as redox cycling of the quinone moiety of DOX and the generation of its toxic metabolites [10,11]. Diverse intercellular and intracellular reactions have been known to produce reactive oxygen species (ROS). These serve as novel cellular signaling molecules, involved in characteristics associated with cellular physiology including growth, differentiation, gene expression, and progression. Moreover, the formation of ROS in excess can result in cell death [12]. The signaling pathways responsible for DOX-induced cardiac apoptosis have received attention from researchers in an effort to gain a deeper understanding. However, in this pathophysiology, the roles of oxidative stress in intracellular signaling pathways and apoptosis have long been established [13,14]. The current study is aimed at examining the protective effect of kirenol against cardiac hypertrophy induced by DOX in H9c2 cells through Nrf2 signaling via PI3K/AKT pathway signaling. HO-1, an endogenous cytoprotective enzyme, is known for its anti-inflammation and anti-apoptotic properties. According to previous studies, the activation of nuclear erythroid-2-related factor (Nrf2) can stimulate HO-1 expression. On the contrary, the deficiency of Nrf2 hinders HO-1 expression, while aggravating oxidative stress both during DOX-induced myocardial hypertrophy and in the heart with transverse aortic constriction [15]. Nrf2 is inactive in the cytoplasm, where it is combined with Keap1, its repressor, under physiological conditions. On the other hand, Nrf2 is released from the Keap1-Nrf2 complex and translocated to the nucleus under conditions of oxidative stress [15,16]. Several studies have shown that nuclear translocation of Nrf2 can be facilitated by the activation of the phosphatidylinositol 3-kinase/protein kinase B (PI3K/AKT) signaling pathway [17,18]. Therefore, this activation and consequent modulation of HO-1 expression may be regarded as a possible approach to hinder cell apoptosis induced by doxorubicin. The activation of c-Jun N-terminal kinases (JNKs) and p38-MAPKs by cellular oxidative stress is also correlated with cardiac pathophysiology and apoptotic cell death. Further, the activation of p38 kinase triggers GATA-dependent B-type natriuretic peptide gene expression. Similarly, the activation of JNK mediates early gene activation that induces hypertrophy-associated transcription factors including c-Jun [19].

Consequently, it is critical to find out the regulatory mechanisms that control Nrf2 gene expression, as the regulation of this gene could provide a better strategy to protect patients against cardiomyocyte death, as well as preventing the progression of heart failure. Several naturally occurring antioxidants have been known to exhibit their effects through the activation of the Nrf2 pathway. In recent studies, it was reported that some natural compounds stimulate Nrf2 nuclear translocation and induce Nrf2-dependent genes that protect cardiomyocytes from toxicity [20,21,22,23]. Kirenol (KRL) is a natural type of diterpenoid extracted from Herba Siegesbeckiae, a historical tropical plant [24]. It has long been used in the treatment of various diseases such as arthritis, malaria, hypertension, and also snakebites, especially in China [24,25,26]. In addition to decreasing the activation of IL-1β, TNF-α, NF-κB, IL-6, and B-cell lymphoma 2 (Bcl-2), kirenol activates the annexin-1, IL-2, BMP, and Wnt pathways. Additionally, this diterpenoid increases LRP-5 and β-catenin mRNA expression, while inhibiting β-catenin-phosphorylating glycogen synthase kinase 3 beta (GSK3β). Attributed to these characteristics, kirenol has been reported to exert anti-inflammatory, antiadipogenic, immunoregulatory, antioxidant, and antiarthritic effects [26,27,28,29,30]. Based on a systematic literature review, KRL may exert an additive effect on cardiac hypertrophy, in addition to reducing oxidative stress and lowering the risk of cardiac diseases [31]. For the present study, H9c2 rat cardiomyocytes were grown under a DOX environment to simulate cardiomyocyte damage. Subsequently, we tested the influence of KRL on the damage. An analysis to determine the role of activated nrf2 signaling via the PI3K/AKT pathway in the cardio-protective effects of KRL was performed.

## 2. Results

### 2.1. KRL Dose-Dependently Inhibits DOX-Induced Cell Death

In the current study, H9c2 cells were incubated with a diverse concentration of KRL (2.5–20 µmol/mL) to clarify the probable toxic effects of the diterpenoid. Then, the cell viability was determined 24 and 48 h after treatment. We found no significant toxic effect on the cell viability following the treatment with KRL for 24 h (Figure 1B,C). However, there was a major decrease in cell viability after treatment with DOX of various concentrations from 24 to 48 h, as compared with the control group (*p* < 0.05) (Figure 1D,E). Figure 1F,G shows that pretreatment with KRL based on a specific time and concentration increases cell viability, following DOX treated for 24 and 48 h (*p* < 0.05 at 15 µmol/mL).

### 2.2. KRL Inhibits Natriuretic Peptide System

Various studies have revealed that the endocrine heart releases ANP and BNP, the cardiac hormones, which play an important role in cardiovascular (CV), renal, and metabolic homeostasis [32,33,34]. These hormones increase in heart failure (HF) as a compensatory homeostatic response to myocardial overload [35]. Consequently, we investigated whether KRL could inhibit the concentration of ANP and BNP in DOX-induced cardiac hypertrophy. We found that the concentration of ANP and BNP significantly increased with DOX 3–3.5 fold as compared to the control group (Figure 2A). On the other hand, the pretreatment of cells with KRL significantly reduced the levels of ANP and BNP. Thus, the current study result demonstrated that KRL has a synergistic effect on cardiac hypertrophy induced by DOX.

### 2.3. KRL Attenuates Activation of Matrix Metalloproteinases 2 and 9

A number of studies suggest that ROS plays a major role in the activation of MMPs; and in this process, the activation of NAD(P)H oxidase, an important source of ROS, is a key event. In both experimental and clinical forms of heart failure, there is an increase in the cardiac content of MMP-2 and MMP-9 [4,36,37,38]. Western blot analysis was performed to determine the ability of KRL to mitigate the induction of MMP-9 and MMP-2. We found that DOX upregulated the levels of MMP-9, MMP-2, and proteins (Figure 2B). On the other hand, KRL reversed this situation, reducing the DOX-induced protein concentration of fibrosis markers at 15 μM. This reduction may be due to the KRL inhibition of the concentrations of MMP-9 and MMP-2.

### 2.4. Effect of KRL on DOX-Induced H9c2 Cytoskeletal Alterations

Earlier experimental studies have remarkably described the role of cytoskeletal alterations of desmin and microtubules in cardiac hypertrophy and failure [39,40].

The cytoskeleton of cardiac myocytes comprises the intermediate filament of desmin, called actin and α- and β-tubulin, which is responsible for the formation of microtubules by polymerization. We observed an accumulation of microtubules, causing an increased load on myocytes, impeding sarcomere motion, and promoting cardiac dysfunction [41,42]. In the current study, we observed that the use of DOX in treating H9c2 cardiomyoblasts led to an expansion of the cell size within 24 hours, as determined from the surface area of cells, indicated by actin filament staining. In pretreatment, KRL-treated cells presented a reduction in the effects of DOX-induced cellular hypertrophy as determined by F-actin staining (Figure 2C). The above results thus validate the ability of KRL to inhibit cardiac hypertrophy induced by DOX in H9c2 cells.

### 2.5. Effect of KRL on DOX-Induced Cell Survival Signaling

Insulin-like growth factor 1 receptor (IG-F1R) is one of the prevalent signaling pathways that regulate survival, proliferation, differentiation, and metabolism of cardiac cells [43,44]. According to previous studies, the PI3K/AKT signaling pathway facilitates an essential cell survival signal in cardiomyocytes [45,46]. Consequently, in the current study, we investigated the role of the IGF-IR-dependent PI3K/AKT pathway in exhibiting the protective effect of exogenous KRL against cardiotoxicity in H9c2 cardiac cells induced by DOX. In DOX-treated H9c2 cells, the expression levels of p-IGF1R, p-PI3K, and p-AKT decreased markedly. However, pretreatment of H9c2 cells with KRL markedly increased the expression levels of these proteins (Figure 3A,B). These results validate the anti-apoptotic effects of KRL, which potentially causes stabilization through the regulation of apoptosis-inducing proteins.

### 2.6. KRL Inhibits Phosphorylation of MAPKs in DOX-Induced Cardiotoxicity

A number of previous studies have assessed the role of p38 mitogen-activated protein kinase (MAPK) in cardiotoxicity induced by DOX. The MAPK family, which includes the extracellular signal-regulated protein kinases p38 kinases, along with the c-Jun N-terminal kinases (JNK), is associated with the processes that promote ROS-mediated cell death [47,48]. Based on the Western blot analysis, p38 and JNK, the two classical MAPKs of cardiac hypertrophy were observed to be activated in the DOX-induced H9c2 cells, indicating their role in cardiac hypertrophy caused by hypertrophy (Figure 3C). The activation of these MAPKs was effectively downregulated upon the treatment of H9c2 cells with KRL. Therefore, our findings indicate that regulating the activation of P38 induces the effect of KRL on hypertrophy.

### 2.7. KRL Suppresses Oxidative Stress in DOX-Induced Cardiomyocytes

Oxidative stress is known to play a key role in cardiomyopathy induced by DOX. ROS and by-products and oxidative stress can induce various types of protein modifications such as aldehydic adducts (4-HNE and MDA), carbonyl derivatives, and 3-NT [49,50]. Consequently, in our study, we performed an analysis using Western blotting to determine whether KRL could inhibit the oxidative markers including 4-HNE and 3-NT. It was found that the DOX treatment of H9c2 cells markedly increased the accumulation of these oxidative markers. Subsequently, KRL treatment prevented this accumulation completely (Figure 4A). This finding demonstrates the probable effects of KRL in suppressing the elevated levels of 4-HNE and 3-NT.

### 2.8. KRL Enhances the Expression of Nrf2 in DOX-Treated H9c2 Cells

To further examine the mechanisms underlying the effects of KRL, we assessed the expression levels of Nrf2 signaling proteins. Figure 4B illustrates an increase in the level of total Nrf2 due to DOX stimulation in cells pretreated with KRL. Furthermore, we observed that exposure to DOX also increased the expression levels of proteins, HO-1, and NQO-1 in KRL pretreatment (Figure 4B). These data showed the implication of Nrf2/HO-1 signaling in the protective effects of KRL against cardiotoxicity induced by DOX.

### 2.9. KRL Upregulates the Translocation of Nrf2

The induction of antioxidant genes in DOX-induced H9c2 cells is assumed to result in the protective effects of KRL. It further mediated the nuclear activation of Nrf2. To emphasize this, we conducted experiments on H9c2 cells in either the absence or presence of DOX. Firstly, the effect of KRL concentration on the expression pattern of nuclear Nrf2 and cytosolic Nrf2 in H9c2 cells was tested. Our Western blot data demonstrated that as compared to the control group, DOX alone and KRL-pretreated, DOX-exposed H9c2 cells showed a dose-dependent rise in the expression of nuclear and cytosolic Nrf2, reaching the maximum at a concentration of 15 μM KRL (Figure 5A). As shown in Figure 5B, this observation remained consistent with the fluorescence microscopy data of this study. It justified the potential antioxidant properties of KRL, exhibiting a protective effect against DOX-induced oxidative stress.

### 2.10. KRL Protects H9c2 Cells Against DOX-Induced Cardiomyocyte Apoptosis

In DOX-induced cardiotoxicity, cardiomyocyte apoptosis remains a key indicator that can also be worsened by oxidative stress and uncontrolled inflammation. The expression of apoptosis-related proteins, Bcl-2 and Bcl-xL, demonstrated DOX-induced apoptosis in H9c2 cells through potential downregulation of the expression levels of these proteins (Figure 6A). KRL treatment, on the other hand, prevented apoptosis by upregulating the proteins for cell survival. Our Western blot analysis also showed changes in the expression of two proteins, cleaved caspase-3 and cleaved PARP, associated with apoptosis. Although DOX upregulated the expression of these proteins, KRL in varying concentrations downregulated the expression levels (Figure 6A). Terminal deoxynucleotidyl transferase dUTP nick-end labeling (TUNEL) staining directly determined the anti-apoptotic capacity of KRL, showing decreased TUNEL-positive nuclei in DOX-treated cells (Figure 6B). These results indicate that KRL inhibited DOX-induced apoptosis.

## 3. Discussion

Herba Siegesbeckiae has historically been used for the treatment of inflammation, hypertension, and arthritis. It is acknowledged that Herba Siegesbeckiae possesses anti-asthma, anticancer, and antibacterial properties. However, its anti-photoaging role remains to be explained [51]. Kirenol is a diterpenoid component which exhibits anti-adipogenesis, antiarthritic, and anti-inflammatory effects found in Herba Siegesbeckiae [51]. Various clinical trials have found that impaired regulation of hyperglycemia, hyperinsulinemia, and diabetic complications such as diabetic cardiomyopathy are closely related. Bin Wu et al. found that kirenol with either 0.5 mg/kg daily dosage or 2.0 mg/kg daily dosage for 8 weeks did not affect bodyweight or lipid profiles in all groups of Goto-Kakizaki rats. Oral administration of kirenol at a daily dose of 2.0 mg/kg decreased fasting plasma glucose levels and fasting plasma insulin by 22–24 weeks of age in Goto-Kakizaki rats and also up to the end of the observation period. HbA1c levels were decreased by kirenol administration only in the high dose kirenol-treated group [51].

Doxorubicin (DOX) is one of the potent anti-cancer drugs that may result in severe cardiotoxicity in cancer patients, posing a serious threat. Non-targeted toxicity has been associated with increased oxidative stress, inflammation, and cell death. The acquired evidence shows that both oxidative stress and cell apoptosis are the key factors for the pathogenesis of DOX-induced myocardial injury, caused by the increased production of ROS (3-NT 4-HNE), leading to cardiomyocyte apoptosis [52,53]. Under multiple pathological events including myocardial infarction, hypertension, and adrenergic over-activation, the myocardium undergoes a hypertrophic response. This is characterized by the expansion of cell size, increased protein synthesis, and stimulation of atrial natriuretic peptide (ANP) and B-type natriuretic peptide (BNP)—fetal cardiac genes [54]. The current study findings confirm the effective role of DOX in inducing the cardiac hypertrophic response, correlated with a rise in the hypertrophic marker proteins ANP and BNP. It is imperative to note that the treatment with KRL promoted effective protection against hypertrophy induced by DOX, with a considerable decline in levels of the above-mentioned proteins in H9c2 cells. As a result, KRL exhibits an efficient protective effect on cardiac hypertrophy.

IGF-1 signaling plays a key role in controlling cell growth, as well as survival. Physiological and adaptive hypertrophic growth of cells induced by the activation of IGF-I via IGF-IR protects the heart from any pathological damage. According to earlier studies, the activation of the IGF-I-PI3K pathway has numerous benefits on cardiac function in chemotherapy-induced cardiotoxicity. When exposed to DOX, there is a decline in the expression levels of IGF-IR, which may be due to DOX-induced oxidative stress. In a rat model of DOX-induced cardiotoxicity, the hyper-activation of PI3K has been observed in the initiation and nucleation steps of autophagy [55,56,57,58]. Additionally, the survival pathways related to IGF-I, including IGF-IR, p-PI3K, and p-AKT, can mediate the cardiac survival pathway. According to our study results, PI3K was markedly downregulated when exposed to DOX. These findings remain consistent with previous results, since the treatment with KRL resulted in the activation of the cardiac survival pathway in cardiomyoblast cells, evidenced by the elevated and restored levels of p-IGF-IR and p-PI3K and p-AKT, respectively.

DOX-induced hypertrophy has been reported to be reliant on MAPK signaling [59,60]. Normally, MAP3 kinase is responsible for the activation of downstream MAPKs that particularly activate JNK and p38 through phosphorylation but not those that activate ERK. Therefore, JNK and p38 may play a crucial role in hypertrophy, involving ROS and mediated by DOX. Cardiomyocytes initiate the MAPK signaling cascade by activating G protein-coupled receptors and stress-induced receptor kinases. The downstream p38 and JNKs phosphorylate an extensive array of intracellular targets and several transcription factors when triggered, leading to the reprogramming of cardiac gene expression [61]. As per our data, KRL-mediated attenuation of hypertrophy associated with hypertrophy/DOX is particularly reliant on the modulation in the P38 MAPK signaling. The transcription factor Nrf2 is responsible for the regulation of essential and inducible expression of a set of antioxidant genes. It is imperative to note that they are ubiquitously expressed in the cardiovascular system. Nrf2 and its downstream targets play a significant role in cardiovascular homeostasis caused by suppressing oxidative stress. They also regulate the onset and progression of heart failure [62]. Nrf2, in particular, controls the transcriptional activation of antioxidant genes by binding with ARE [62]. Despite several studies validating the antioxidant properties of KRL, its underlying mechanism has not been elucidated in detail. In H9c2 cells, we observed that KRL increased the Nrf2-regulated genes such as HO-1 and NQO-1, which was prevented by cells treated with DOX alone. Thus, it indicated that the activation of Nrf2 remains crucial for KRL-mediated antioxidant and pro-survival effects.

Anti-apoptotic proteins such as Bcl-2 and Bcl-xL not only inhibit the release of cytochrome c, but also protect the cardiac cells from oxidative stress and apoptosis [44]. On the contrary, caspase-3 executes the apoptotic program and is primarily responsible for the cleavage of PARP at the time of cell death [63]. Additionally, the cleavage of PARP-1 in fragments of 89 and 24 kDa becomes a significant hallmark of this type of cell death during apoptosis. This cleavage is well studied and produced by caspases 3 and 7, which are activated during apoptosis. Our study findings showed that DOX-induced downregulation of Bcl2 and Bcl-xL and the elevation of cleaved caspase-3 and PARP cleavage were upregulated and suppressed by KRL treatment in H9c2 cells, respectively. The KRL treatment, however, suppressed the activation of the caspase cascade. Our results showed a concomitant decrease in cellular antioxidant status of H9c2 cardiomyocytes with overwhelming ROS. The toxic effects of doxorubicin on cardiac cells has been previously reported. DOX exposure to these cells promotes apoptosis by suppressing IGF-1R, PI3K/AKT signaling, and the Nrf2 pathway [31].

## 4. Materials and Methods

### 4.1. Reagents

For our sample, we purchased doxorubicin (D1515) and kirenol (52659-56-0) from Sigma-Aldrich Co. (St. Louis, MO, USA). A terminal deoxynucleotidyl transferase dUTP nick-end labeling (TUNEL) staining kit (Roche, cat. no. 11684817910), antibodies against pAKT (# 44-621G), AKT (# 44-609G), pPI3K (#PA5-104853), PI3K (# PA5-29220), BCL2 (# PA5-27094), PARP (PA5-16452), HO-1 (# PA5-77833), NQO1 (# PA5-82294), pNrf2 (# PA5-105664), β-actin (# PA5-78716), and Lamin B1 (# PA5-19468), ANP (# 711569), MMP9 (PA5-13199), MMP2 (PA5-85197), and rhodamine phalloidin (R415) were sourced from Invitrogen, Thermo Fisher Scientific, Inc. (Waltham, MA, USA). Furthermore, we purchased anti-cleaved caspase-3 (ab32042) and Anti-BNP antibody (ab19645) from Abcam (Branford, CT, USA) and Lipofectamine 2000 (11668027) and Nrf2 siRNA from Thermo Fisher Scientific, Inc. (Waltham, MA, USA). We also procured 3-nitrotyrosine (3-NT, 1:3000 dilutions, Millipore, Billerica, MA), 4-hydroxy-2-nonenal (4-HNE, Alpha Diagnostic International, San Antonio, TX, USA), and plasminogen activator inhibitor-1 (PAI-1, BD Biosciences, San Jose, CA, USA).

### 4.2. Cell Culture

We obtained H9c2 cardiomyoblasts from the American Type Culture Collection (ATCC, Manassas, VA, USA) and cultured them in Dulbecco’s modified essential medium (DMEM, ThermoFisher Scientific, Waltham, MA, USA). This was supplemented with 10% fetal bovine serum (ThermoFisher Scientific), 2 mM glutamine (Sigma-Aldrich, St. Louis, MO, USA), 100 unit/mL penicillin (Sigma-Aldrich), 100 μg/mL streptomycin (Sigma-Aldrich), and 1 mM pyruvate (Sigma-Aldrich) in humidified air (5% CO_2_) at 37 °C. To investigate the protective effect of KRL against cellular hypertrophy, we treated H9c2 cells with 25 µM (EC50) of KRL for 2 hours before the doxorubicin challenge. After 24 h, we collected the cell lysate for analysis.

### 4.3. Cell Viability

The H9c2 cells were passaged and cultured in 24-well plates at 5 × 104 cells/well for a day. Subsequently, the cells were treated with KRL for 2 h prior to adding doxorubicin. After 24 h, the trypan blue exclusion assay was used to determine the percentage of cell death under a light microscope, as mentioned above. Then the viable cell ratio was calculated as follows: viable cell ratio (%) = (unstained cell number/total cell number) × 100.

### 4.4. Preparation of Cytoplasmic and Nuclear Extracts

For the preparation of cytoplasmic extracts, the cell pellets were resuspended in Buffer I, consisting of 25 mM HEPES pH 7.9, 5 mM KCl, 0.5 mM MgCl2, and 1 mM dithiothreitol (DTT), for 5 min. Then, we mixed this suspension with the same amount of Buffer II, which consisted of 25 mM HEPES pH 7.9, 5 mM KCl, 0.5 mM MgCl2, and 1 mM DTT. Additionally, we mixed the suspension with protease and phosphatase inhibitors supplemented with 0.4% (*v*/*v*) NP40. The obtained suspension samples were incubated at 4 °C for 15 min with rotation, followed by the centrifugation of lysates in a microfuge at 2500 rpm at 4 °C for 5 min. The supernatants were then transferred to new Eppendorf tubes. The cell pellets were cleaned with Buffer II once, and we added the supernatant to the cytoplasmic protein tube. The lysates were centrifuged again at 4 °C at 10,000× *g* for 5 min for the removal of residual nuclei, and they were transferred to new Eppendorf tubes. For nuclear extraction, we subjected the pellets obtained from cytoplasmic extraction to incubation with Buffer III, consisting of HEPES 25 mM, pH of 7.9, NaCl—400 mM, 10 percent of sucrose or dextrose, 0.05% NP-40, and 1 mM DTT, in addition to protease and phosphatase inhibitors. The lysates were rotated at 4 °C for 1 h and centrifuged at 4 °C at 1000 rpm for 10 min. Following this step, nuclear proteins were able to be found in the collected supernatants.

### 4.5. Western Blotting

After the treatment, the cells were harvested and washed once with cold PBS. This process was followed by the preparation of cytoplasmic, nuclear, and total extracts in the aforementioned manner. For the detection of the protein status, a Bio-Rad protein assay was used in each sample, with bovine serum albumin (BSA) as the reference standard. To resolve the equal amounts of protein (50 μg), we used SDS-PAGE (8–15%) and transferred the proteins to nitrocellulose membranes overnight. The membranes were blocked using 5% skimmed milk at 3 °C for 30 min, followed by their incubation with the indicated primary antibodies for 2 h. Subsequently, a horseradish peroxidase-conjugated goat anti-mouse or anti-rabbit secondary antibody was incubated with the nitrocellulose membranes for 1 h. An enhanced chemiluminescence substrate (Pierce Biotechnology, Rockford, IL, USA) and an LI-COR chemiluminescence imaging system (3600-00-C-Digit Blot Scanner) were used to develop the membranes and examine the samples, respectively. We used Image Studio Lite software (LI-COR Biosciences, Lincoln, NE, USA) to generate the graphs of the intensities of the densitometric band with normalization to the intensity of the untreated control band set to 1.

### 4.6. Phalloidin Staining

As described in our previous report with slight modification, the cells were stained with actin staining for the analysis of the hypertrophic effects on H9c2 cells (54). The cells were briefly cultured to 70% confluence in chambered slides (SPL life sciences, Pocheon-si, Korea) and stained with rhodamine phalloidin stain.

### 4.7. Immunostaining

Post 24-h administration of KRL and/or DOX, the cells were fixed with 4% paraformaldehyde in 1 × PBS at room temperature (RT) for 15 min and permeabilized with 0.1% Triton X-100 in 0.1% sodium citrate for 20 min. To detect the cell nuclei (blue staining), the cells were stained with rhodamine-conjugated phalloidin (Life Technologies, Carlsbad, CA, USA) for 15 min and 1 μg/mL DAPI for 5 min, after blocking with 5% bovine serum albumin. For the visualization of Nrf2, the cells were incubated with anti-Nrf2 antibody (diluted in 1% BSA solution) for 12 h after blocking, and further with Alexa Fluor 488 goat anti-mouse antibody for 1 h. After washing the cells in PBS, they were stained with DAPI.

### 4.8. TUNEL Assay

At the logarithmic growth stage, H9c2 cells were loaded in and supplemented with a six-well plate and KRL or DOX, respectively. Then, after removing the medium, the cells were cleaned with PBS and processed with 4% paraformaldehyde for 20 min, followed by the removal of paraformaldehyde. Again, PBS was used to re-wash the cells, which were then incubated with TUNEL reagent (11684817910, Roche, Mannheim, Germany). After washing the cells with PBS, they were counterstained with 0.1 μg/mL DAPI for 5 min and examined using a fluorescence microscope. We executed all the morphometric studies three times. TUNEL-positive cells were detected as brilliant green, whereas the cell nuclei were observed through UV light microscopy at 454 nm. Images were obtained through microscopy (200× magnification), and we used a Leica D6000 fluorescence microscope for measurement (Leica, Wetzlar, Germany).

### 4.9. Statistical Analysis

We used GraphPad Prism software version 6.0 (GraphPad Software Inc., San Diego, CA, USA) and one-way ANOVA for statistical analyses and the comparison of three groups. We represented the data as the mean ± SD, and *p* < 0.05 was considered significant.

## 5. Conclusions

Demonstrating the inhibition of cardiac hypertrophy by KRL in vitro, our study shows the potential cardio-protective effects of KRL. However, further investigation of the molecular pathway is required to validate this hypothesis. Extensive future work is also required to narrow down the mechanism underlying KRL’s protective effects on myocytes. More time-interval-related studies and in-depth probing into apoptosis are likely to provide a clear picture to a considerable extent. We acknowledge that our study fails to provide an explanation for the expression levels of the apoptosis-related proteins. A deeper understanding of these protein levels could have provided us with an alternative perspective on our entire project. Further work in vivo on suitable SHR animal models will aid us in clarifying the effects of KRL on models of cardiotoxicity induced by DOX.

## Figures and Tables

**Figure 1 ijms-22-03269-f001:**
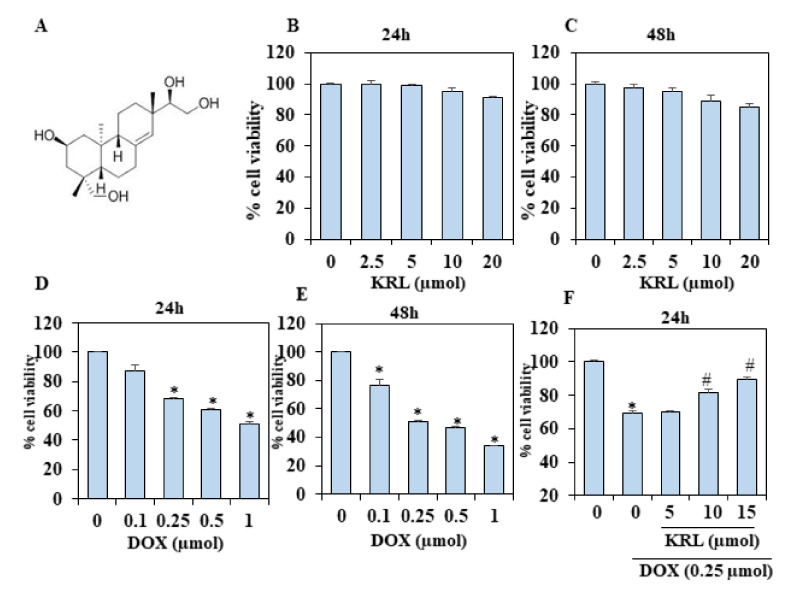
Kirenol (KRL)-attenuated DOX-induced cytotoxicity in H9c2 cardiac cells. (**A**) Chemical structure of kirenol. (**B**,**C**) Effect of KRL on the viability of H9c2 cardiomyocytes examined by an MTT assay. H9c2 cells were treated with increasing concentrations of KRL (2.5–20 μmol) for 24 and 48 h, respectively. Cell viability (%) was measured as follows: (A570 of treated cells/A570 of untreated cells) × 100. (**D**,**E**) The cell viability with increasing concentrations of DOX (0.1–1 μmol) for 24 and 48 h, respectively. (**F**,**G**) The effects of KRL along with DOX on the cell viability of H9c2 cells were determined by MTT assay. H9c2 cells were cultured in serum free media for 3 h followed by the treatment with KRL for 2 h before or after DOX treatment, respectively. Data are represented as the mean ± SD of triplicate values (*n* = 3) and * *p* < 0.05 represents significant variations compared with the control. # *p* < 0.05 represents significant variations as compared to DOX alone and KRL with DOX treatment groups.

**Figure 2 ijms-22-03269-f002:**
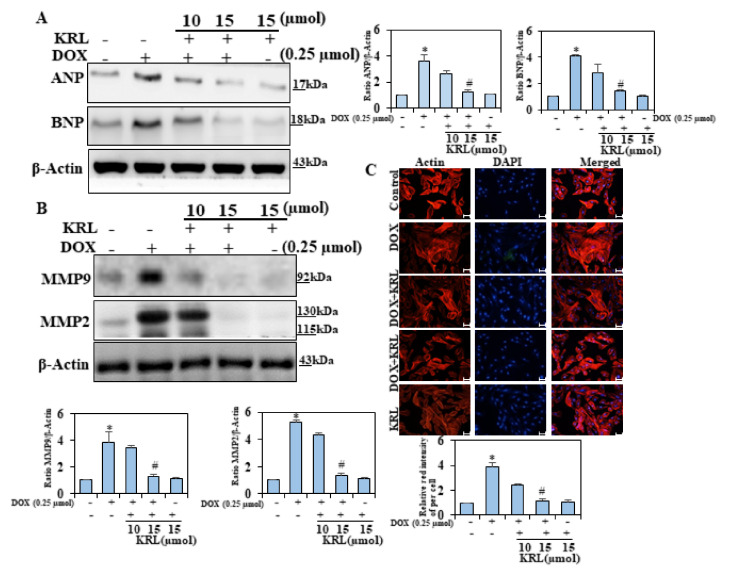
Effects of KRL on DOX-induced hypertrophy. H9c2 cells were treated for 24 h with 15 µM of KRL, 2 h before DOX (0.25 µmol) treatment. (**A**) Representative Western blots showing changes in the protein levels of ANP and BNP. (**B**) KRL is illustrated to downregulate MMP9 and MMP2. Western blot analysis was performed to determine the total protein MMP9 and MMp2 levels in the total extract by including β-actin as an internal loading control. (**C**) H9c2 cells were treated for 24 h with 10 and 15 µM of KRL, 2 h before the DOX (0.25 µmol) treatment. The cells then underwent actin filament staining to observe the changes in the surface area of the cardiomyocytes. Scale bar indicated 100µ m at 20× magnification. Data are represented as the mean ± SD of triplicate values (*n* = 3) and * *p* < 0.05 represents significant variations compared with the control. # *p* < 0.05 represents significant variations as compared to DOX alone and KRL with DOX treatment groups.

**Figure 3 ijms-22-03269-f003:**
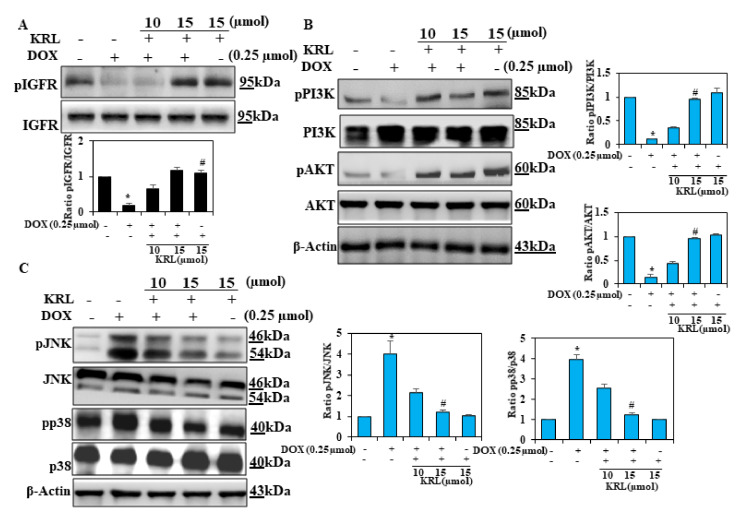
KRL enhances cell survival mechanism. H9c2 cells were cultured in serum-free media for 3 h, followed by treatment with KRL for 2 h before or after DOX treatment, respectively. (**A**,**B**) KRL activates the IGF1R-mediated survival pathway in H9c2 cells. The expression of p-IGF1R, PI3K, p-PI3K, AKT, and p-AKT was analyzed by Western blotting. β-actin was used as the internal control. (**C**) Representative Western blots showing the changes in MAPK signaling proteins (pP38 and pJNK) in H9c2 cells. Data are represented as the mean ± SD of triplicate values (*n* = 3) and * *p* < 0.05 represents significant variations compared with the control. # *p* < 0.05 represents significant variations as compared to DOX alone and KRL with DOX treatment groups.

**Figure 4 ijms-22-03269-f004:**
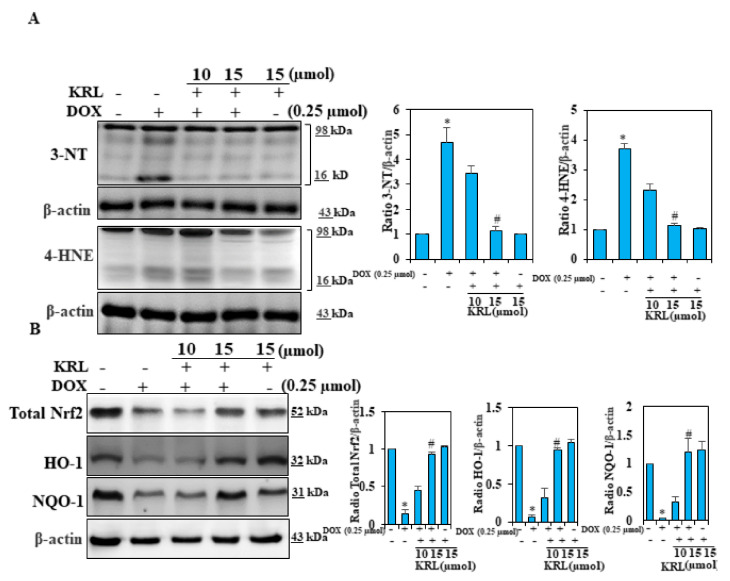
KRL prevents DOX-induced cardiac oxidative stress. (**A**) Cells were treated with KRL (15 μmol) for 2 h prior to DOX (0.25 µmol) stimulation for 24 h. Changes in 3-nitrotyrosine (3-NT) and 4-hydroxy-2-nonenal (4-HNE) protein levels of mitochondria were monitored by Western blotting. (**B**) The activation of total Nrf2, HO-1, and NQO-1 was examined by Western blotting. Data are represented as the mean ± SD of triplicate values (*n* = 3) and * *p* < 0.05 represents significant variations compared with the control. # *p* < 0.05 represents significant variations as compared to DOX alone and KRL with DOX treatment groups.

**Figure 5 ijms-22-03269-f005:**
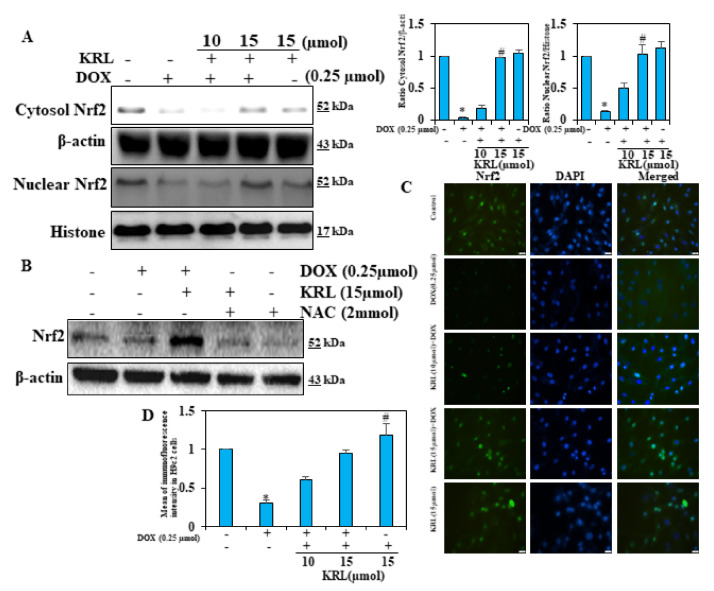
KRL enhances NRf2 translocation in DOX-treated H9c2 cells. (**A**) Changes in Nrf2 levels were determined in nuclear and cytosolic fractions of cells. Relative changes in protein intensities were quantified using Image Studio Lite software and presented as a histogram, with the control set at one-fold. (**B**) Cells subjected to reactive oxygen species (ROS) inhibitor treatment (1 mM NAC for 60 min) before KRL treatment for 24 h. The expression of total Nrf2 was analyzed by Western blotting. (**C**) Cells were pre-treated with KRL (15 μmol) for 2 h and then stimulated with or without DOX (0.25 µmol) for 24 h. Immunofluorescence staining was performed to detect the nuclear localization of Nrf2. Following incubation with primary antibody (anti-Nrf2) and conjugated secondary antibody, cells were stained with DAPI (1 μg/mL) for 5 min. The subcellular localization of Nrf2 in all conditions was visualized under fluorescence microscopy. Scale bar indicated 100µ m at 20× magnification. (**D**) Indicated Nrf2 immunofluorescence intensity levels of Control and treated H9c2 cells. Data are represented as the mean ± SD of triplicate values (*n* = 3) and * *p* < 0.05 represents significant variations compared with the control. # *p* < 0.05 represents significant variations as compared to DOX alone and KRL with DOX treatment groups.

**Figure 6 ijms-22-03269-f006:**
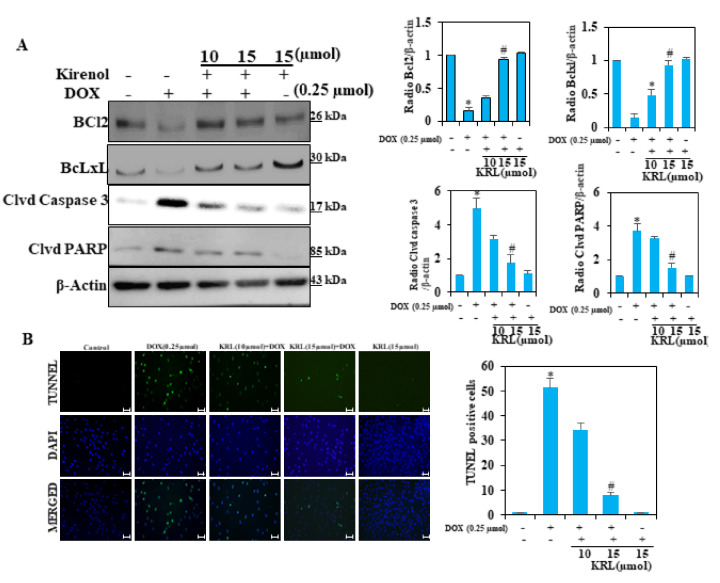
KRL inhibits DOX-induced apoptosis in cardiomyocytes. (**A**) Cells were pretreated with 15 µmol of KRL for 2 h followed by DOX (0.25 µmol for 24 h). The expression levels of anti-apoptotic Bcl2 and Bcl-xL and apoptotic proteins cleaved caspase-3 activation and PARP cleavage were measured by the Western blot method. (**B**) Apoptotic nuclei were detected by terminal deoxynucleotidyl transferase dUTP nick-end labeling (TUNEL) staining and the nuclei were detected by DAPI staining, showing modulations in apoptotic levels with DOX against and with KRL. Scale bar indicated 100 µm at 20× magnification. Data are represented as the mean ± SD of triplicate values (*n* = 3) and * *p* < 0.05 represents significant variations compared with the control. # *p* < 0.05 represents significant variations as compared to DOX alone and KRL with DOX treatment groups.

## Data Availability

The data that support the findings of this study are available from the corresponding author upon reasonable request.

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
