# Peer review of "Cardiac Protective Effect of Kirenol against Doxorubicin-Induced Cardiac Hypertrophy in H9c2 Cells through Nrf2 Signaling via PI3K/AKT Pathways"

_ijms, 2021, doi:10.3390/ijms22063269_

Round 1
Reviewer 1 Report
Doxorubicin is a chemotherapeutic used to treat a range of cancers and heart failure is one of the most common complications of its use. Many studies have examined how DOX damages cardiomyocytes through increased oxidative damage, and tested the ability of a range of antioxidants and natural health products to attenuate doxorubicin-induced oxidative stress and apoptosis. Previous studies have investigated the anti-tumour activity of combined kirenol and doxorubicin. In this study the authors investigate the potential for cardioprotective effects of kirenol against apoptosis in H9c2 cells in combination with doxorubicin. The findings are novel though they rely on a single cell line and do not show that any of the effects of KRL are present in primary cardiomyocytes or human cells. Moreover, the authors do not definitively determine the mechanism responsible for kirinol’s activity and represent an incremental advance of knowledge in this field.
Major comments
- 1: KRL appears to marginally reduce cell viability at high dosages over 48h, as does DOX. Does KRL also attenuate DOX-induced cytotoxicity over 48h of treatment?
- 1: It is unclear whether the experiment shown in panel F, is performed by pretreating with KRL or treating the cells with KRL and DOX at the same time.
- Figure 2: please show quantitation of data in panel C.
- A number of immunoblots are overexposed and could be improved. These include PI3K in panel B, and pp38/p38 in panel C. Moreover the immunoblots are not representative of the quantitative graphed data shown in the panels (B and C), especially for the 10microM KRL dosage.
- Figure 3: how were the cells treated? Were they pre-treated with KRL? And for how long? (describe this in lines 155-160).
- The source of the ROS production in the H9c2 cells is unclear. Does KRL protect against mitochondrial ROS or cytosolic ROS production? Or both?
- The authors do not have
- Figure 4: The authors imply that the effects of KRL are dependent upon restoring Nrf2 expression, but the authors should provide evidence that the effects of KRL are dependent upon Nrf2. For example using siRNA to knockdown Nrf2, does this abolish the beneficial effects of KRL?
- Figure 5 seems to show that both nuclear and cytosolic Nrf2 are similarly affected by DOX and KRL. However, Nrf2 should be normalized to an exclusively nuclear protein, not GAPDH.
- How does KRL affect Nrf2 levels? Is it transcriptional, translational or is it regulating the stability of the Nrf protein?
- Figures 5B and 6B should be quantified.
- The discussion does not consider the limitations of the research design or discuss how the dosages chosen are relevant to the anticipated systemic bioavailability of KRL.
Minor comments
Line 66 describes the purpose of the study to be the protective effect of kirenol against “cardiac hypertension”. This should be changed to cardiac hypertrophy.
Line 96 refers to a systematic review, please refence this publication.
Line 138: What is the DOX II challenge? Please indicate the DOX dosage used in the legend for the panel C experiment.
Author Response
Reviewer 1
Thank you for giving us the opportunity to submit a revised draft of our manuscript entitled “Cardiac protective effect of kirenol against doxorubicin-induced cardiac hypertension in H9c2 cells through Nrf2 signaling via PI3K/AKT pathways” [Manuscript ID: IJMS 1116895] to International Journal of Molecular Science.
We appreciate the time and effort that you and the reviewers have dedicated to providing valuable feedback on our manuscript. We are grateful to the reviewers for their insightful comments, which have improved the paper.
We have incorporated changes that reflect all the suggestions provided by the reviewers. All changes are highlighted in green in the revised manuscript. Please see below for our point-by point response to the reviewers’ comments. If any responses are unclear or you wish additional changes, please do not hesitate to let us know.
1: KRL appears to marginally reduce cell viability at high dosages over 48h, as does DOX. Does KRL also attenuate DOX-induced cytotoxicity over 48h of treatment?
Thank you for your valuable comments: In our study, we had discussed KRL being considered as a potential agent for hypertension resulting from cardiac deterioration and hypertrophy and not for apoptosis. KRL attenuates DOX-induced cytotoxicity over 48h of treatment which is evidenced in our study. We had done the experiment earlier and the results are attached along with this revision.
1: It is unclear whether the experiment shown in panel F, is performed by pretreating with KRL or treating the cells with KRL and DOX at the same time.
Thank you for your comments: H9c2 cells were cultured in serum free media for 3 h followed by the treatment with KRL for 2 h before or after DOX treatment respectively. Our revised manuscript we have indicated the cells treatment. (Line 117)
Figure 2: please show quantitation of data in panel C.
In our revised manuscript, quantified data was added as suggested.
Figure 3
A number of immunoblots are overexposed and could be improved. These include PI3K in panel B, and pp38/p38 in panel C. Moreover the immunoblots are not representative of the quantitative graphed data shown in the panels (B and C), especially for the 10microM KRL dosage.
Thank you for your comments. The figure 3B was redesigned. Pp38 and p38 we have done new blot and our revised manuscript modified with the above-mentioned data
Figure 3: how were the cells treated? Were they pre-treated with KRL? And for how long? (describe this in lines 155-160).
We made the correction as per your suggestion.
The source of the ROS production in the H9c2 cells is unclear. Does KRL protect against mitochondrial ROS or cytosolic ROS production? Or both?
ROS source from Mitochondria. Our revised manuscript has been described. H9c2 cells were cultured in serum free media for 3 h followed by the treatment with KRL for 2 h before or after DOX treatment respectively and incubate 24 h, mitochondria were prepared, and Western blot analysis was performed with antibodies recognizing 3-NT and 4-HNE protein adducts.
Figure 4: The authors imply that the effects of KRL are dependent upon restoring Nrf2 expression, but the authors should provide evidence that the effects of KRL are dependent upon Nrf2. For example using siRNA to knockdown Nrf2, does this abolish the beneficial effects of KRL?
Thanks for the comment. Unfortunately, we do not have sufficient stock of lipofectamine. we have ordered lipofectamine, but the suppliers said there are no guarantee due to corona pandemic. We are very sorry; we will consider your suggestions in future. At the same time, our revised manuscript have demonstrated that with KRL, increases the activity of Nrf2 by using NAC ( ROS inhibitor). Our revised manuscript this data has been added to our revised manuscript (Fig 5B). Thank you
Figure 5 seems to show that both nuclear and cytosolic Nrf2 are similarly affected by DOX and KRL. However, Nrf2 should be normalized to an exclusively nuclear protein, not GAPDH.
Thank you for your comments. As suggested, in our revised manuscript, we have modified the above mentioned data. We have normalised to an exclusively nuclear protein Histone instead of GAPDH.
How does KRL affect Nrf2 levels? Is it transcriptional, translational or is it regulating the stability of the Nrf protein?
KRL regulate the Nrf2 signaling by post-translational, transcriptional and most of the diterpenoid compounds regulate the Nrf2 signaling by both. We have to plan to study the regulation of Nrf2 by KRL at molecular level. (Complete activation of Nrf2 pathway by the synthesized hybrids)
Figures 5B and 6B should be quantified.
In our revised manuscript, fluorescence intensity figure was added as suggested.
The discussion does not consider the limitations of the research design or discuss how the dosages chosen are relevant to the anticipated systemic bioavailability of KRL.
Thank you for your valuable comments: in our revised manuscript, we have described as you suggested
Minor comments
Line 66 describes the purpose of the study to be the protective effect of kirenol against “cardiac hypertension”. This should be changed to cardiac hypertrophy.
We made the correction as per your suggestion.
Line 96 refers to a systematic review, please refence this publication.
We made the correction as per your suggestion
Line 138: What is the DOX II challenge? Please indicate the DOX dosage used in the legend for the panel C experiment.
Typographical error we made corrected and indicated DOX dosage.
Thank you for all your valuable comments and questions, which allowed us to improve the quality of the manuscript.
Reviewer 2 Report
In the present study, the authors investigate the investigate the cardioprotective effects of a biologically active substance extracted from S. orientalis, Kirenol (KRL), against apoptosis in H9c2 cells induced by doxorubicin. They show the beneficial effects of KRL in preventing DOX-induced cardiac oxidative stress, remodeling and apoptosis. Mechanistically, they demonstrate that KRL not only activates the IGF-IR- dependent p-PI3K/p-AKT and Nrf2 signaling pathway but also suppresses the caspase-dependent apoptosis. Although the manuscript is interesting, there are several concerns that should be addressed. Listed below some specific comments.
- As far as I am concerned, for such a manuscript title the Authors would need an in vivo model to evaluate doxorubicin-induced cardiac hypertension and determine the beneficial effects of kirenol in the context of an animal (for example SHR animals as stated by the Authors).
- Figures are completely not sharp. Please enhance dpi.
- Figure 6 B. The authors show the antiapoptotic capacity of kirenol on DOX‐treated cardiomyocytes through TUNEL staining assay. Consider adding a graph with calculation of cardiomyocyte apoptotic index.
- Lines 296-297: Please correct MAPKK in the sentence.
- Immunofluorescence figures lack of scale bars. Please add. In addition, the unit of measurement for the scale bar should be included in the figure legend for each panel.
- In the Materials and methods section the Authors describe immunostaining methodologies for GATA4 visualization after 24-hour administration of KRL and/or DOX. However, these data were not included in the manuscript. Please justify this incongruence.
- Lines 324-325, the Authors stated “Additionally, DOX‐exposure of these cells promotes apoptosis by suppressing IGF‐1R, PI3K/AKT signaling, and the Nrf2 pathway”. For such a claim, it appears that the Authors would highlight the previously reported toxic effects of doxorubicin on cardiac cells (DOI: 10.1002/jcb.26305; DOI: 10.1007/BF03349163).
- The authors should include a scheme that visualizes the presented major findings.
Author Response
In the present study, the authors investigate the investigate the cardioprotective effects of a biologically active substance extracted from S. orientalis, Kirenol (KRL), against apoptosis in H9c2 cells induced by doxorubicin. They show the beneficial effects of KRL in preventing DOX-induced cardiac oxidative stress, remodeling and apoptosis. Mechanistically, they demonstrate that KRL not only activates the IGF-IR- dependent p-PI3K/p-AKT and Nrf2 signaling pathway but also suppresses the caspase-dependent apoptosis. Although the manuscript is interesting, there are several concerns that should be addressed. Listed below some specific comments.
Thank you for giving us the opportunity to submit a revised draft of our manuscript entitled “Cardiac protective effect of kirenol against doxorubicin-induced cardiac hypertension in H9c2 cells through Nrf2 signaling via PI3K/AKT pathways” [Manuscript ID: IJMS 1116895] to International Journal of Molecular Science.
We appreciate the time and effort that you and the reviewers have dedicated to providing valuable feedback on our manuscript. We are grateful to the reviewers for their insightful comments, which have improved the paper.
We have incorporated changes that reflect all the suggestions provided by the reviewers. All changes are highlighted in green in the revised manuscript. Please see below for our point-by point response to the reviewers’ comments. If any responses are unclear or you wish additional changes, please do not hesitate to let us know.
As far as I am concerned, for such a manuscript title the Authors would need an in vivo model to evaluate doxorubicin-induced cardiac hypertension and determine the beneficial effects of kirenol in the context of an animal (for example SHR animals as stated by the Authors).
Thank you for your valuable comments. One year before we need to order the SHR animals from Charles River Laboratories, unfortunately we did not receive the animals due to pandemic diseases. We hope that in coming June we expect the animals, after that we will continue further research in In Vivo model.
Figures are completely not sharp. Please enhance dpi.
Thank you for your comments, our revised manuscript show enhanced DPI resolution and attached figures only in Zip file, but in manuscript word file, quality of figures not sharp.
Figure 6 B. The authors show the antiapoptotic capacity of kirenol on DOX‐treated cardiomyocytes through TUNEL staining assay. Consider adding a graph with calculation of cardiomyocyte apoptotic index.
In our revised manuscript, we have added graph according your suggestion
Lines 296-297: Please correct MAPKK in the sentence.
We made the correction as per your suggestion.
Immunofluorescence figures lack of scale bars. Please add. In addition, the unit of measurement for the scale bar should be included in the figure legend for each panel.
In our revised manuscript, we have added scale bar according your suggestion
In the Materials and methods section the Authors describe immunostaining methodologies for GATA4 visualization after 24-hour administration of KRL and/or DOX. However, these data were not included in the manuscript. Please justify this incongruence.
Typographical error, we made correction in our revised manuscript (line 415)
Lines 324-325, the Authors stated “Additionally, DOX‐exposure of these cells promotes apoptosis by suppressing IGF‐1R, PI3K/AKT signaling, and the Nrf2 pathway”. For such a claim, it appears that the Authors would highlight the previously reported toxic effects of doxorubicin on cardiac cells (DOI: 10.1002/jcb.26305; DOI: 10.1007/BF03349163).
We made the correction as per your suggestion. (Line 347)
The authors should include a scheme that visualizes the presented major findings.
As per your suggestion, we have included graphical abstract in our revised manuscript
Thank you for all your valuable comments and questions, which allowed us to improve the quality of the manuscript.
Round 2
Reviewer 1 Report
The authors have appropriately addressed the comments of the reviewer. However the grammatical errors need to be corrected before the manuscript can be published. For example, in the figure 4 legend, correct to "mitochondria".
In addition, the first paragraph of the discussion needs to be fixed. It does not read logically and the sentences are too long and have grammatical errors.
Author Response
Reviewer 1
Thank you for giving us the second opportunity to submit a revised draft of our manuscript entitled “Cardiac protective effect of kirenol against doxorubicin-induced cardiac hypertension in H9c2 cells through Nrf2 signaling via PI3K/AKT pathways” [Manuscript ID: IJMS 1116895] to International Journal of Molecular Science. Please see below for our point-by point response to the reviewers’ comments. If any responses are unclear or you wish additional changes, please do not hesitate to let us know.
- The authors have appropriately addressed the comments of the reviewer. However the grammatical errors need to be corrected before the manuscript can be published. For example, in the figure 4 legend, correct to "mitochondria".
Thank you for your valuable comments, we have done our revised manuscript as you suggested.
In addition, the first paragraph of the discussion needs to be fixed. It does not read logically and the sentences are too long and have grammatical errors.
Thank you for your comments, we have done our revised manuscript as you suggested.

Reviewer 2 Report
Thank the Authors for the diligently and thoroughly revised version of your ms.
Author Response
Thank you very much for your comments